# QUANTIZATION-AWARE DIFFUSION MODELS FOR MAXIMUM LIKELIHOOD TRAINING

**Shohei Taniguchi, Masahiro Suzuki & Yutaka Matsuo**
Department of Technology Management for Innovation
The University of Tokyo
7-3-1 Hongo, Bunkyo-ku, Tokyo, Japan
{taniguchi,masa,matsuo}@weblab.t.u-tokyo.ac.jp

## ABSTRACT

Diffusion models are powerful generative models for continuous signals, such as images and videos. However, real-world digital data are quantized; hence, they take not continuous values but only a finite set of discrete values. For example, pixels in 8-bit images can take only 256 discrete values. In existing diffusion models, quantization is either ignored by treating data as continuous, or handled by adding small noise to make the data continuous. Neither approach guarantees that samples from the model will converge to the finite set of quantized points. In this work, we propose a methodology to explicitly account for quantization within diffusion models. Specifically, by adopting a particular form of parameterization, we guarantee that samples from the reverse diffusion process converge to quantized points. In experiments, we demonstrate that our quantization-aware model can substantially improve the performance of diffusion models for density estimation, and achieve state-of-the-art results on pixel-level image generation in likelihood evaluation. In particular, for CIFAR-10 image generation, the negative log-likelihood improves substantially from 2.42 to 0.27, approaching the theoretical lower bound.

## 1 INTRODUCTION

Diffusion probabilistic models (Sohl-Dickstein et al., 2015; Ho et al., 2020) and score-based generative models (Song & Ermon, 2019; 2020) have achieved state-of-the-art performance in terms of sample quality and density estimation (Kingma et al., 2021; Sahoo et al., 2024) for image generation. Both models consider to perturb data with a sequence of noise distributions, and generate samples by learning to reverse the diffusion process from noise to data. Song et al. (2020b) have shown that these two types of models can be interpreted using a single framework, which we refer to as *diffusion models* in this paper.

The framework of diffusion models (Song et al., 2020b) involves gradually diffusing the data distribution towards a simple noise distribution, such as a Gaussian distribution, using a stochastic differential equation (SDE), and learning the time reversal of this SDE for generative modeling. The reverse-time SDE has an analytic expression which only depends on a time-dependent score function of the perturbed data distribution. This score function can be efficiently estimated by training a neural network (called a score-based model (Song & Ermon, 2019; 2020)) with a weighted combination of score matching losses (Hyvärinen & Dayan, 2005; Vincent, 2011; Song et al., 2020a) as the objective. After training, we can obtain samples from the model by simulating the reverse SDE from a simple noise using the estimated score function.

By definition, diffusion models are designed for continuous-valued data, and have been applied to continuous domains (e.g., images (Croitoru et al., 2023), videos (Ho et al., 2022b;a), and audio (Zhang et al., 2023; Huang et al., 2023)). However, even in such domains, digital data do not take continuous values in a precise sense due to quantization. For example, a pixel in 8-bit images can take only an integer value from 0 to 255 rather than a real continuous value. In many previous works of diffusion models, quantization is largely ignored during training, since its effect seems negligible to the performance, and they only apply it to generated samples during inference as post-

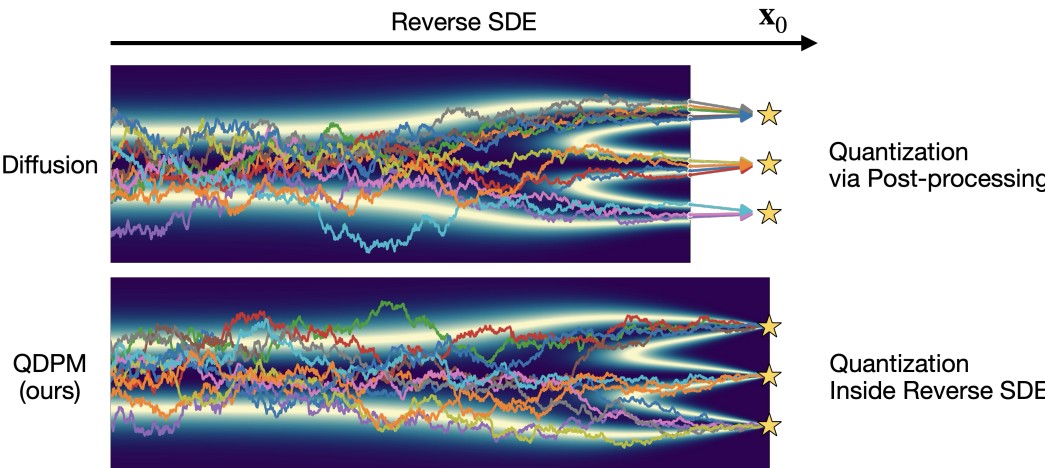

Figure 1: Overview of our QDPM. In the traditional diffusion models (above), the reverse SDE does not converge to quantized points the data lie on, so quantization is performed as a post-processing after SDE simulation. On the other hand, our QDPM (below) incorporates the quantization process inside the reverse SDE, which explicitly guarantee convergence to the quantized point without post-processing.

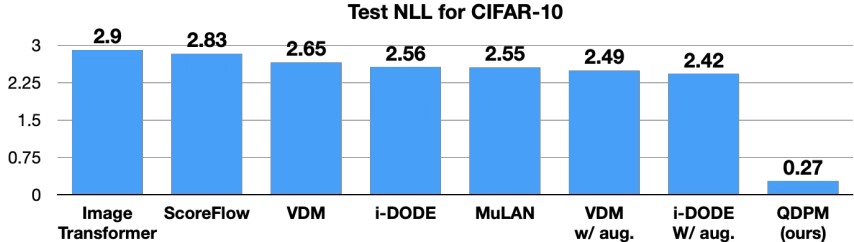

Figure 2: Comparison of the test negative log-likelihood of our QDPM for CIFAR-10 with existing methods. QDPM achieves the state-of-the-art performance by a large margin.

processing (Karras et al., 2022). In some studies, they deal with quantization by adding small noise to quantized data before applying them to the diffusion model so that the data lie on a continuous space. However, such preprocessing leads to the trained model generating noisy data that may deteriorate the performance.

In this paper, we propose a method that directly incorporates quantization into the design of score-based diffusion models, which can discard such ad hoc pre- or post-processing. Specifically, we derive a sufficient condition for the reverse-time SDE to converge to a quantized value as shown in Figure 1, and propose a specific parameterization to meet the condition. We refer to the model trained under this framework as the Quantizing Diffusion Probabilistic Model (QDPM). In the experiments, we show that our QDPM can substantially improve the performance of pixel-level image generation. In particular, QDPM achieves a negative log-likelihood of 0.27 bpd on CIFAR-10, which beats the current state-of-the-art result of 2.42 by i-DODE (Zheng et al., 2023) by a large margin.

## 2 BACKGROUND

### 2.1 DIFFUSION MODELS

Diffusion models (Song et al., 2020b) are generative models defined via a stochastic differential equation (SDE) known as the diffusion process. For example, letting the data be $\mathbf{x}_0 \in \mathbb{R}^d$, consider the following variance-exploding (VE) SDE.

$$\mathrm{d}\mathbf{x}_t = \sqrt{2t}\,\mathrm{d}\mathbf{w} \tag{1}$$

Let $q_t$ and $q_{0t}$ denote the marginal distribution at time $t$ and the transition density of this SDE, respectively.

$$q_{0t}\left(\mathbf{x}_t \mid \mathbf{x}_0\right) = \mathcal{N}\left(\mathbf{x}_t; \mathbf{x}_0, t^2\boldsymbol{I}\right) \tag{2}$$

The time-reverse process of this forward SDE can be analytically derived as follows:

$$\mathrm{d}\mathbf{x}_t = -2t \cdot \mathbf{s}_t\left(\mathbf{x}_t\right)dt + \sqrt{2t}\,\mathrm{d}\bar{\mathbf{w}}, \tag{3}$$

where $s_t$ is the score function of the noisy data distribution $q_t$ at time $t$, i.e., $\mathbf{s}_t(\mathbf{x}_t) = \nabla_{\mathbf{x}_t}\log q_t(\mathbf{x}_t)$. If we can simulate this reverse process, we can model the generative process of the data distribution. However, the score function is unknown and must be estimated. To that end, we consider a model $\hat{\mathbf{s}}_\theta\left(\mathbf{x}_t, t\right)$ that estimates the score function and train it so that the estimated reverse SDE matches the true SDE.

$$\mathrm{d}\mathbf{x}_t = -2t \cdot \hat{\mathbf{s}}_\theta\left(\mathbf{x}_t, t\right)\mathrm{d}t + \sqrt{2t}\,\mathrm{d}\bar{\mathbf{w}} \tag{4}$$

Let $p_t$ denote the marginal distribution at time $t$ for this (estimated) reverse SDE. Since it suffices that the scores match at any time $t$, we minimize the following score-matching loss.

$$\mathcal{J}_{\mathrm{SM}}\left(\theta\right) = \mathbb{E}_{\mathbf{x}_0 \sim p_{\mathrm{data}}}\left[\int_{t_{\min}}^{t_{\max}} w\left(t\right)\left\|\mathbf{s}_t\left(\mathbf{x}_t\right) - \hat{\mathbf{s}}_\theta\left(\mathbf{x}_t, t\right)\right\|^2 dt\right], \tag{5}$$

where $w(t)$ is a positive weighting function over time, i.e., $w(t) > 0$, $0 \le t_{\min} < t_{\max} < \infty$, and $\|\cdot\|$ denotes the $L^2$-norm. Because $\mathbf{s}_t$ is unknown, we cannot compute this directly; however, it is known that minimizing $\mathcal{J}_{\mathrm{SM}}$ is equivalent to minimizing the denoising score-matching loss below.

$$\mathcal{J}_{\mathrm{DSM}}\left(\theta\right) = \mathbb{E}_{\mathbf{x}_0 \sim p_{\mathrm{data}},\,\boldsymbol{\epsilon} \sim \mathcal{N}(\mathbf{0}, \boldsymbol{I})}\left[\int_{t_{\min}}^{t_{\max}} w\left(t\right)\left\|-\frac{\boldsymbol{\epsilon}}{t} - \hat{\mathbf{s}}_\theta\left(\mathbf{x}_t = \mathbf{x}_0 + t \cdot \boldsymbol{\epsilon}, t\right)\right\|^2 dt\right] \tag{6}$$

$$= \mathbb{E}_{\mathbf{x}_0 \sim p_{\mathrm{data}},\,\boldsymbol{\epsilon} \sim \mathcal{N}(\mathbf{0}, \boldsymbol{I})}\left[\int_{t_{\min}}^{t_{\max}} \frac{w\left(t\right)}{t^2}\left\|\boldsymbol{\epsilon} - \hat{\boldsymbol{\epsilon}}_\theta\left(\mathbf{x}_t = \mathbf{x}_0 + t \cdot \boldsymbol{\epsilon}, t\right)\right\|^2 dt\right] \tag{7}$$

$$= \mathcal{J}_{\mathrm{SM}}\left(\theta\right) + \mathrm{const.} \tag{8}$$

Here we set $\hat{\boldsymbol{\epsilon}}_\theta = -t \cdot \hat{\mathbf{s}}_\theta$. In other words, if we train a model $\hat{\boldsymbol{\epsilon}}_\theta$ to predict the added noise $\boldsymbol{\epsilon}$ from noisy data $\mathbf{x}_t$, we can effectively estimate the underlying score function. If, instead of a noise predictor, we define a signal predictor as $\hat{\mathbf{x}}_\theta = \mathbf{x}_t - t \cdot \hat{\boldsymbol{\epsilon}}_\theta$, then $\mathcal{J}_{\mathrm{DSM}}$ can be rewritten as follows:

$$\mathcal{J}_{\mathrm{DSM}}\left(\theta\right) = \mathbb{E}_{\mathbf{x}_0 \sim p_{\mathrm{data}},\,\boldsymbol{\epsilon} \sim \mathcal{N}(\mathbf{0}, \boldsymbol{I})}\left[\int_{t_{\min}}^{t_{\max}} \frac{w\left(t\right)}{t^4}\left\|\mathbf{x}_0 - \hat{\mathbf{x}}_\theta\left(\mathbf{x}_t = \mathbf{x}_0 + t \cdot \boldsymbol{\epsilon}, t\right)\right\|^2 dt\right] \tag{9}$$

Furthermore, when $w(t) = t$, it is known that minimizing $\mathcal{J}_{\mathrm{SM}}$ corresponds to maximizing a lower bound on the likelihood of $p_0$.

$$-\log p_0\left(\mathbf{x}_0; \theta\right) \le \mathbb{E}_{\boldsymbol{\epsilon} \sim \mathcal{N}(\mathbf{0}, \boldsymbol{I})}\left[\int_{t_{\min}}^{t_{\max}} \frac{1}{t^3}\left\|\mathbf{x}_0 - \hat{\mathbf{x}}_\theta\left(\mathbf{x}_t = \mathbf{x}_0 + t \cdot \boldsymbol{\epsilon}\right)\right\|^2 dt\right] + c_1 + c_2 \tag{10}$$

$$c_1 = -\mathbb{E}_{\boldsymbol{\epsilon} \sim \mathcal{N}(\mathbf{0}, \boldsymbol{I})}\left[\log p_\theta\left(\mathbf{x}_0 \mid \mathbf{x}_{t_{\min}} = \mathbf{x}_0 + t_{\min} \cdot \boldsymbol{\epsilon}\right)\right], \quad c_2 = \frac{1}{2t_{\max}^2}\left\|\mathbf{x}^{(i)}\right\|_2^2 \tag{11}$$

Up to this point we considered a VE-SDE, but by applying a change of variables to $\mathbf{x}_t$ one can derive analogous forms for other SDEs such as variance-preserving (VP) and straight-through (ST) SDEs. For example, if we set $\mathbf{y}_t = \mathbf{x}_t/\sqrt{1 + t^2}$ the process follows a VP-SDE, while $\mathbf{y}_t = \mathbf{x}_t/(1 + t)$ yields an ST-SDE.

## 2.2 Dequantization for Diffusion models

Standard diffusion models assume that the data $\mathbf{x}_0$ take continuous values. In contrast, real-world digital data such as images are quantized and therefore, take only a finite set of values rather than true continuous values. This quantization is ignored in many prior works, which typically treat the data directly as continuous (Karras et al., 2022) or use diffusion models in the continuous latent space in the VAE (Rombach et al., 2022; Peebles & Xie, 2023).

When one wants to evaluate models by likelihood, however, it is common to handle quantization explicitly in order to compare against quantization-aware models such as autoregressive models (e.g., PixelCNN (Van Den Oord et al., 2016; Salimans et al., 2017)). The simplest and most common approach is to treat the data with small added noise as $\mathbf{x}_0$ and evaluate the likelihood of the quantized data via a variational bound (Song et al., 2021; Zheng et al., 2023). For example, with 8-bit images whose pixel values are integers in $0 \sim 255$, one often first adds i.i.d. uniform noise in $[0, 1)$ to the pixel values to make them continuous. As a related alternative, one sets the start time of the diffusion process $t_{\min}$ to a small positive value and defines $p_\theta\left(\mathbf{x}_0 \mid \mathbf{x}_{t_{\min}}\right)$ as a probability distribution over the quantized discrete space (Kingma et al., 2021; Sahoo et al., 2024). In this case, $p_\theta\left(\mathbf{x}_0 \mid \mathbf{x}_{t_{\min}}\right)$ becomes a categorical distribution:

$$p_\theta\left(\mathbf{x}_0 \mid \mathbf{x}_{t_{\min}}\right) = \prod_i^d p_\theta\left((\mathbf{x}_0)_i \mid \mathbf{x}_{t_{\min}}\right), \tag{12}$$

$$p_\theta\left((\mathbf{x}_0)_i = x^{(k)} \mid \mathbf{x}_{t_{\min}}\right) = \operatorname*{softmax}_k\left(-\frac{\left(x^{(k)} - (\mathbf{x}_{t_{\min}})_i\right)^2}{2t_{\min}^2}\right), \tag{13}$$

where we suppose each element of the data (e.g., each RGB pixel) is quantized to a value in $\left\{x^{(k)}\right\}_{k=1}^K$. In the case of an 8-bit image pixel, $K = 256$ and $x^{(k)} = k$.[1] While these methods are convenient for likelihood evaluation, they are ad-hoc and do not guarantee that the estimated reverse SDE will converge to quantized points.

# 3 Method: Quantizing Diffusion Probabilistic Models

We propose a methodology that designs a quantization-aware score function so that the reverse SDE explicitly converges to quantized points; we call this approach Quantizing Diffusion Probabilistic Models (QDPM). To guarantee this, we first analyze the solution of the reverse SDE and derive sufficient conditions. Based on it, we then design a score function that satisfies those conditions.

## 3.1 Quantization-Aware Parameterization

Here we consider the case where a signal predictor $\hat{\mathbf{x}}_\theta$ is parameterized. Given an initial condition $\mathbf{x}_s$, the solution for $\mathbf{x}_t$ at time $t$ of the reverse SDE for $0 \le t \le s < \infty$ can be derived as follows:

$$\mathbf{x}_t = \frac{t^2}{s^2}\mathbf{x}_s - t^2 \int_s^t \frac{2}{r^3}\hat{\mathbf{x}}_\theta\left(\mathbf{x}_r, r\right)\mathrm{d}r + t^2 \int_s^t \sqrt{\frac{2}{r^3}}\mathrm{d}\mathbf{w}_r. \tag{14}$$

See the appendix for the derivation. For the reverse SDE to converge to a quantized point, we have to ensure that the limit of this solution at $t \to 0$ always lies at a quantized point. As regards the limit point $\mathbf{x}_0$, the following fact holds under some regularity conditions:

**Proposition 1** (Convergence to a fixed point, informal). *For the stochastic differential equation defined by Eq. (14), the solution $\mathbf{x}_0$ at $t = 0$ meets the following equality almost surely.*

$$\mathbf{x}_0 = \hat{\mathbf{x}}_\theta\left(\mathbf{x}_0, 0\right). \tag{15}$$

This means that the limit point $\mathbf{x}_0$ is always a fixed point of the signal predictor $\hat{\mathbf{x}}_\theta\left(\cdot, 0\right)$. To verify whether this holds in practice, we measure the fixed-point error, i.e., $\|\mathbf{x}_t - \hat{\mathbf{x}}_\theta(\mathbf{x}_t, t)\|^2/d$, at each time step $t$ when sampling from the pretrained EDM (Karras et al., 2022) on CIFAR-10 and AFHQ

---

[1]Note that, in many cases, pixel values are scaled and shifted to a standardized range (e.g., $[-1, 1]$) rather than raw indices via preprocessing, but we omit it for simplicity here.

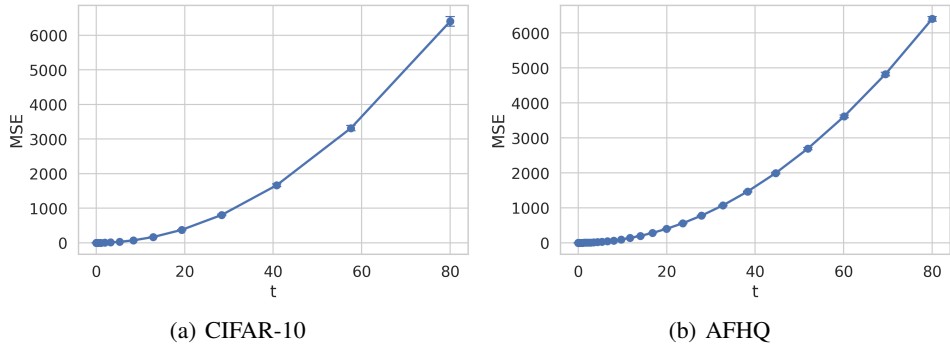

(a) CIFAR-10            (b) AFHQ

Figure 3: Fixed-point error of the pretrained EDM on CIFAR-10 and AFHQ at sampling steps. The error bars show the standard deviation of 64 samples.

and plot it in Figure 3. As can be seen from it, the fixed-point error converges to zero as $t \to 0$, confirming that Proposition 1 holds true in practice. Therefore, by limiting the fixed points of the signal predictor $\hat{\mathbf{x}}_\theta$ to quantized points, we can guarantee that the solution of the reverse SDE at $t \to 0$ lies on them. Based on this fact, we can derive a sufficient condition for the SDE to converge to quantized points as follows:

**Proposition 2** (Sufficient condition for convergence to a quantized point, informal)**.** *If the signal predictor $\hat{\mathbf{x}}_\theta$ meets the following equality for any $\mathbf{x} \in \mathbb{R}^d$, the solution $\mathbf{x}_0$ at $t = 0$ takes a value in a set of quantized points, i.e., $\Omega = \left( \left\{ x^{(k)} \right\}_{k=1}^K \right)^d$, almost surely.*

$$\hat{\mathbf{x}}_\theta \left( \mathbf{x}, 0 \right) = \operatorname{round} \left( \mathbf{x} \right) := \underset{\mathbf{y} \in \Omega}{\arg\min} \left\| \mathbf{x} - \mathbf{y} \right\|. \tag{16}$$

When this condition is satisfied, it is obvious that the limit point $\mathbf{x}_0$ always lies at a quantized point due to Eq. (15). In fact, we can construct a signal predictor $\hat{\mathbf{x}}_\theta$ that satisfies the above condition as follows:

$$\hat{\mathbf{x}}_\theta \left( \mathbf{x}_t, t \right) = \operatorname{softround} \left( \mathbf{x}_t, t \right) + e^{-1/t^2} \hat{\boldsymbol{\delta}}_\theta \left( \mathbf{x}_t,\ t \right), \tag{17}$$

$$\left( \operatorname{softround} \left( \mathbf{x}_t, t \right) \right)_i = \sum_{k=1}^K \operatorname{softmax} \left( -\frac{\left( x^{(k)} - \left( \mathbf{x}_t \right)_i \right)^2}{2t^2} \right) \cdot x^{(k)}, \tag{18}$$

where $(\cdot)_i$ is the $i$-th element of a vector, and $\hat{\boldsymbol{\delta}}_\theta$ is some parametric function, e.g., a neural network. Intuitively, $\operatorname{softround}(\cdot)$ is a smooth version of $\operatorname{round}(\cdot)$ and converges to it as $t \to 0$. Since the second term in Eq. (17) diminishes as $t \to 0$, the sufficient condition is always satisfied under this parameterization.

## 3.2 Maximum Likelihood Loss for QDPM

As described in Section 2.1, the negative log-likelihood of diffusion models admits the upper bound in Eq. (10). Taking the limits $t_{\min} \to 0$ and $t_{\max} \to \infty$ yields $c_1, c_2 \to 0$, so we obtain the following (negative) evidence lower bound (ELBO):

$$- \log p_0 \left( \mathbf{x}_0; \theta \right) \le \mathbb{E}_{\boldsymbol{\epsilon} \sim \mathcal{N}(\mathbf{0}, \boldsymbol{I})} \left[ \int_0^\infty \frac{1}{t^3} \left\| \mathbf{x}_0 - \hat{\mathbf{x}}_\theta \left( \mathbf{x}_t = \mathbf{x}_0 + t \cdot \boldsymbol{\epsilon} \right) \right\|^2 \mathrm{d}t \right] = \mathcal{L} \left( \mathbf{x}_0, \theta \right) \tag{19}$$

If we omit the terms in $\mathcal{L}$ that are independent of the model $\hat{\boldsymbol{\delta}}_\theta$, we can rewrite the loss as follows:

$$\mathcal{L} \left( \mathbf{x}_0, \theta \right) = \mathbb{E} \left[ \int_0^\infty \frac{e^{-1/t^2}}{t^3} \hat{\boldsymbol{\delta}}_\theta^\top \left( e^{-1/t^2} \hat{\boldsymbol{\delta}}_\theta - 2\boldsymbol{\delta}_t \right) \mathrm{d}t \right] + c, \tag{20}$$

$$\text{where } \boldsymbol{\delta}_t = \mathbf{x}_0 - \operatorname{softround} \left( \mathbf{x}_t, t \right), \ c = \mathbb{E} \left[ \int_0^\infty \frac{\left\| \boldsymbol{\delta}_t \right\|^2}{t^3} \mathrm{d}t \right]. \tag{21}$$

---

**Algorithm 1** QDPM-Solver-1

---

$\mathbf{x}_s,\ u_s \leftarrow \mathbf{0},\ 1$
**for** $i \leftarrow 1$ to $M$ **do**
$\quad u_t \leftarrow u_s - 1/M$
$\quad s,\ t \leftarrow 1/\sqrt{-\log u_s},\ 1/\sqrt{-\log u_t}$
$\quad \hat{\mathbf{x}}_\theta\left(\mathbf{x}_s, s\right) = \text{softround}\left(\mathbf{x}_s, s\right) + u_s \cdot \hat{\boldsymbol{\delta}}_\theta\left(\mathbf{x}_s,\ s\right)$
$\quad \mathbf{x}_t \sim \mathcal{N}\left(\frac{t^2}{s^2}\mathbf{x}_s + \left(1 - \frac{t^2}{s^2}\right)\hat{\mathbf{x}}_\theta\left(\mathbf{x}_s, s\right),\ t^2\left(1 - \frac{t^2}{s^2}\right)\boldsymbol{I}\right)$
$\quad u_s \leftarrow u_t,\ \mathbf{x}_s \leftarrow \mathbf{x}_t$
**end for**
**return** $\mathbf{x}_t$

---

With the change of variable $u = e^{-1/t^2}$, this can be further written in a simple form:

$$\mathcal{L}\left(\mathbf{x}_0, \theta\right) = \mathbb{E}\left[\int_0^\infty \frac{e^{-1/t^2}}{t^3}\hat{\boldsymbol{\delta}}_\theta^\top\left(e^{-1/t^2}\hat{\boldsymbol{\delta}}_\theta - 2\boldsymbol{\delta}_t\right)\mathrm{d}t\right] + c \tag{22}$$

$$= \mathbb{E}\left[\int_0^\infty \hat{\boldsymbol{\delta}}_\theta^\top\left(\frac{e^{-1/t^2}}{2}\hat{\boldsymbol{\delta}}_\theta - \boldsymbol{\delta}_t\right)\frac{2e^{-1/t^2}}{t^3}\mathrm{d}t\right] + c \tag{23}$$

$$= \mathbb{E}\left[\int_0^1 \hat{\boldsymbol{\delta}}_\theta^\top\left(\frac{u}{2}\hat{\boldsymbol{\delta}}_\theta - \boldsymbol{\delta}_t\right)\mathrm{d}u\right] + c = \mathbb{E}\left[\hat{\boldsymbol{\delta}}_\theta^\top\left(\frac{u}{2}\hat{\boldsymbol{\delta}}_\theta - \boldsymbol{\delta}_t\right)\right] + c, \tag{24}$$

where $u \sim \mathcal{U}\left(0, 1\right),\ t = 1/\sqrt{-\log u},\ \mathrm{d}u = \frac{2e^{-1/t^2}}{t^3}\mathrm{d}t$. Therefore, we can train the quantization-aware diffusion model with this maximum likelihood objective.

Regarding the design of $\hat{\boldsymbol{\delta}}_\theta$, we use a U-Net-type architecture following the prior works (Ho et al., 2020; Kingma et al., 2021; Karras et al., 2022). However, since our formulation is based on VE-SDE, the noised data $\mathbf{x}_t$ can take a large value especially when $t$ is large. To normalize the magnitude, we input $\mathbf{x}_t/\sqrt{1 + t^2}$ to the neural net instead of the raw $\mathbf{x}_t$. Similarly, we also use $u = e^{-1/t^2}$ as an input instead of $t$, since $u$ has a bounded range of $[0, 1]$ whereas $t$ is unbounded.

## 3.3 EFFICIENT SDE SOLVER FOR QDPM

Since we already have the exact solution for the reverse SDE in Eq. (14), we can construct an efficient solver rather than using a naive Euler–Maruyama solver. The simplest way is to use the first-order approximation for the drift term as follows:

$$\mathbf{x}_t = \frac{t^2}{s^2}\mathbf{x}_s - t^2\int_s^t \frac{2}{r^3}\hat{\mathbf{x}}_\theta\left(\mathbf{x}_r, r\right)\mathrm{d}r + t^2\int_s^t \sqrt{\frac{2}{r^3}}\mathrm{d}\mathbf{w}_r \tag{25}$$

$$\approx \frac{t^2}{s^2}\mathbf{x}_s - t^2\int_s^t \frac{2}{r^3}\hat{\mathbf{x}}_\theta\left(\mathbf{x}_s, s\right)\mathrm{d}r + t^2\int_s^t \sqrt{\frac{2}{r^3}}\mathrm{d}\mathbf{w}_r \tag{26}$$

$$\sim \mathcal{N}\left(\frac{t^2}{s^2}\mathbf{x}_s + \left(1 - \frac{t^2}{s^2}\right)\hat{\mathbf{x}}_\theta\left(\mathbf{x}_s, s\right),\ t^2\left(1 - \frac{t^2}{s^2}\right)\boldsymbol{I}\right). \tag{27}$$

By running it for discretized points in $t \in [0, \infty)$, we can simulate the reverse SDE. Since the time variable $t$ can take a value in an unbounded range, which is difficult to discretize, we choose the discretized points in the space of $u = e^{-1/t^2} \in [0, 1]$, and run the update of $\mathbf{x}_t$ by transform it into $t$. The algorithm of this QDPM-Solver is summarized in Algorithm 1. We can also derive a second-order alternative of this solver using the scheme of Runge–Kutta methods as shown in Algorithm 2.

In fact, our derived SDE solver is closely related to DPM-Solver++ (Lu et al., 2022b). DPM-Solver++ is also derived using a closed-form solution using the signal predictor $\hat{\mathbf{x}}_\theta$, but its formulation is based on VP-SDE instead of VE-SDE. In addition, DPM-Solver++ chooses the discretization points in the space of $\lambda = -\log t$, whereas we take them in the space of $u = e^{-1/t^2}$. In our QDPM setting, $t$ can take a value in $[0, \infty)$, so $\lambda$-based discretization is not feasible because $\lambda$ also takes a value in an unbounded range $(-\infty, \infty)$.

---

**Algorithm 2** QDPM-Solver-2

$\mathbf{x}_s, u_s \leftarrow \mathbf{0}, 1$
**for** $i \leftarrow 1$ to $M$ **do**
    $u_r, u_t \leftarrow u_s - 2/(3M), u_s - 1/M$
    $s, r, t \leftarrow 1/\sqrt{-\log u_s}, 1/\sqrt{-\log u_r}, 1/\sqrt{-\log u_t}$
    $\boldsymbol{\xi}_r \sim \mathcal{N}\left(\mathbf{0}, r^2\left(1 - \frac{r^2}{s^2}\right)\boldsymbol{I}\right), \boldsymbol{\xi}_t \sim \mathcal{N}\left(\frac{t^2}{r^2}\boldsymbol{\xi}_r, t^2\left(1 - \frac{t^2}{r^2}\right)\boldsymbol{I}\right)$
    $\hat{\mathbf{x}}_\theta\left(\mathbf{x}_s, s\right) = \text{softround}\left(\mathbf{x}_s, s\right) + u_s \cdot \hat{\boldsymbol{\delta}}_\theta\left(\mathbf{x}_s, s\right)$
    $\mathbf{x}_r \leftarrow \frac{r^2}{s^2}\mathbf{x}_s + \left(1 - \frac{r^2}{s^2}\right)\hat{\mathbf{x}}_\theta\left(\mathbf{x}_s, s\right) + \boldsymbol{\xi}_r$
    $\hat{\mathbf{x}}_\theta\left(\mathbf{x}_r, r\right) = \text{softround}\left(\mathbf{x}_r, r\right) + u_r \cdot \hat{\boldsymbol{\delta}}_\theta\left(\mathbf{x}_r, r\right)$
    $\mathbf{x}_t \leftarrow \frac{t^2}{s^2}\mathbf{x}_s + \left(1 - \frac{t^2}{s^2}\right)\left(\frac{1}{4}\hat{\mathbf{x}}_\theta\left(\mathbf{x}_s, s\right) + \frac{3}{4}\hat{\mathbf{x}}_\theta\left(\mathbf{x}_r, r\right)\right) + \boldsymbol{\xi}_t$
    $u_s \leftarrow u_t, \mathbf{x}_s \leftarrow \mathbf{x}_t$
**end for**
**return** $\mathbf{x}_t$

---

# 4 RELATED WORK

## 4.1 DEALING WITH DATA QUANTIZATION IN DIFFUSION MODELS

Although many existing works ignore the data quantization, and treat the raw data as a continuous signal, some studies take it into account for their model design, especially when they want to perform the likelihood-based evaluation. For example, the original DDPM (Ho et al., 2020) uses a discritized Gaussian distribution for $p_\theta\left(\mathbf{x}_0 \mid \mathbf{x}_{t_{\min}}\right)$, and explicitly quantize the output of the diffusion model, even though they use a discrete-time model instead of continuous $t$. VDM (Kingma et al., 2021) also explicitly deals with the quantization by simply defining $p_\theta\left(\mathbf{x}_0 \mid \mathbf{x}_{t_{\min}}\right)$ as a categorical distribution over the quantized points as in Eq. (13). Some works (Song et al., 2020b; 2021) use a technique called *dequantization*, in which a small continuous noise is added to data before passing it to the models, and the likelihood of the original quantized data is calculated as a variational bound. For example, with 8-bit images whose pixel values are integers in $0 \sim 255$, one often first adds i.i.d. uniform noise in $[0, 1)$ to the pixel values to make them continuous. In this case, the negative log-likelihood (NLL) can be upper bounded using Jensen's inequality as follows:

$$-\log p\left(\mathbf{x}; \theta\right) = -\log \int_{[\mathbf{0}, \mathbf{1})^d} p\left(\tilde{\mathbf{x}} = \mathbf{x} + \mathbf{u}; \theta\right) d\mathbf{u} \leq -\mathbb{E}_{\mathbf{u} \sim \mathcal{U}(\mathbf{0}, \mathbf{1})}\left[\log p\left(\tilde{\mathbf{x}} = \mathbf{x} + \mathbf{u}; \theta\right)\right]. \quad (28)$$

In many cases, the dequantized data is scaled and shifted in a range $[-1, 1]$; hence, by defining $\mathbf{y} = \tilde{\mathbf{x}}/128 - 1$, we can treat the (continuous) NLL of $\mathbf{y}$ as an upper bound of the NLL of the original quantized data $\mathbf{x}$ as follows:

$$-\log p\left(\mathbf{x}; \theta\right) \leq -\mathbb{E}_{\mathbf{u} \sim \mathcal{U}(\mathbf{0}, \mathbf{1})}\left[\log p\left(\tilde{\mathbf{x}} = \mathbf{x} + \mathbf{u}; \theta\right)\right] \quad (29)$$

$$= -\mathbb{E}_{\mathbf{u} \sim \mathcal{U}(\mathbf{0}, \mathbf{1})}\left[\log p\left(\mathbf{y} = \frac{\mathbf{x} + \mathbf{u}}{128} - 1; \theta\right) + \log\left|\det \frac{d\mathbf{y}}{d\tilde{\mathbf{x}}}\right|\right] \quad (30)$$

$$= -\mathbb{E}_{\mathbf{u} \sim \mathcal{U}(\mathbf{0}, \mathbf{1})}\left[\log p\left(\mathbf{y} = \frac{\mathbf{x} + \mathbf{u}}{128} - 1; \theta\right)\right] + 7d. \quad (31)$$

The dequantization method is not only applied to diffusion models but also to any generative models for continuous-valued data, such as normalizing flows (Ho et al., 2019). More sophisticated noise, such as truncated Gaussians, rather than uniform noise is often used for dequantization (Zheng et al., 2023; Song et al., 2021) in prior works. In contrast to these works, our QDPM incorporates the quantization process inside the design of reverse SDE, which can eliminate any ad hoc operations from both training and inference.

## 4.2 MAXIMUM LIKELIHOOD TRAINING OF DIFFUSION MODELS

Originally, Ho et al. (2020) derived a maximum likelihood objective for the discrete-time diffusion model, but they experimentally show that a simpler noise prediction loss performs better in terms of the sample quality. After Song et al. (2020b) reformulate the continuous-time diffusion model using

stochastic differential equations, Song et al. (2021) and Huang et al. (2021) derive the corresponding ELBO objective for it. In previous works of maximum likelihood training for continuous-time diffusion models, the time variable $t$ is typically truncated within a predefined bounded region in $[t_{\min}, t_{\max}]$, because training tends to suffer from numerical instability, otherwise. We assume that such numerical instability is related to the lack of quantization awareness of diffusion models in prior work. As can be seen from the ELBO objective in Eq. (10), the upper bound of negative log-likelihood can easily diverge if $t_{\min}$ approaches zero, since the coefficient $t^{-3}$ will be divergent. Therefore, training with the ELBO objective will be intractable when we want to set a small $t_{\min}$. Kim et al. (2022) try to alleviate this numerical instability by introducing a technique called *soft truncation*, in which the truncation time $t_{\min}$ is randomly chosen during training. Although soft truncation alleviates the numerical instability during training, it still requires the choice of a minimum truncation time $t_{\min}$. On the other hand, our QDPM parameterization in Eq. (17) ensures that the signal predictor $\hat{\mathbf{x}}_\theta$ approaches to the original quantized signal $\mathbf{x}_0$ as $t \to 0$ at super-exponential rate by definition. Hence, the divergent behavior of the ELBO objective can be avoided, making maximum likelihood training feasible without time truncation.

## 4.3 PARAMETERIZATION OF DIFFUSION MODELS

In the original paper by Song et al. (2020b), the noise predictor $\hat{\epsilon}_{\boldsymbol{\theta}}$ is directly parameterized by a neural network (e.g., U-Net), and many subsequent works follow that parameterization. However, some variants are also proposed in the previous works, such as the signal predictor (Karras et al., 2022; Wei et al., 2023), the velocity predictor (Salimans & Ho, 2022; Zheng et al., 2023). In our QDPM parameterization, we can interpret that the function $\hat{\boldsymbol{\delta}}_\theta$ predicts $\boldsymbol{\delta}_t$, which is the difference between the data signal $\mathbf{x}_0$ and the smoothly rounded value of the current state, i.e., $\mathrm{softround}\,(\mathbf{x}_t, t)$, with a scale of $u$, since the loss function in Eq. (24) can be rewritten as follows:

$$\mathcal{L}\left(\mathbf{x}_0, \theta\right) = \mathbb{E}\left[\frac{1}{2u}\left\|u \cdot \hat{\boldsymbol{\delta}}_\theta - \boldsymbol{\delta}_t\right\|^2\right]. \tag{32}$$

This quantization-aware parameterization contributes to a drastic improvement in density estimation for quantized signal data (e.g., images) as described in the next section.

## 4.4 SDE SOLVERS FOR DIFFUSION MODELS

Originally, Song et al. (2020b) use a hybrid approach that combines the Euler–Maruyama solver and Langevin dynamics, which they call *predictor-corrector sampling*. In subsequent studies, more sophisticated approaches have been proposed. Karras et al. (2022) uses a second-order solver based on Heun's method to decrease the discretization errors. Lu et al. (2022b) proposes a SDE solver named DPM-Solver based on the exact solution of the reverse SDE, which is similar to QDPM solver except for the choice of discretization points. Although we focus on SDE solvers in this paper, ODE solvers for diffusion models (and flow matching) have been proposed in many prior works (Karras et al., 2022; Lu et al., 2022a). We leave the derivation of an ODE alternative of our QDPM-Solver for future work.

## 5 EXPERIMENTS

**Datasets:** To demonstrate the effectiveness of our QDPM, we perform experiments of image generation and density estimation tasks. In our experiment, we use the CIFAR-10 and downsampled ImageNet-32 (Deng et al., 2009) datasets for density estimation. In some previous works (Parmar et al., 2018; Chen et al., 2018; Song et al., 2021), the old version of ImageNet-32 is used, which is no longer available, so we clarify it in the table. We also train QDPM for Flickr-Faces-HQ (FFHQ), and Animal-Faces-HQ (AFHQ) datasets with $64 \times 64$ resolution for qualitative evaluation. We do not employ any data augmentation during training, since we observe that QDPM can achieve good density estimation results without it. Regarding the model architecture, we use the U-Net-based model used in Kingma et al. (2021), which is one of the most commonly used models for image density estimation (Zheng et al., 2023; Sahoo et al., 2024).

**Evaluation:** We evaluate the model performance with the negative log-likelihood (NLL) of the reverse SDE. Following prior works, we report the bits-per-dimension (BPD) score, which corre-

Table 1: Density estimation performance on CIFAR-10 and ImageNet-32. Negative log-likelihood (NLL) in bits per dimension (BPD) is reported. * Corresponds to the old version of ImageNet-32.

| Model | CIFAR-10 | ImageNet |
|---|---|---|
| Image Transformer (Parmar et al., 2018) | 2.90 | 3.77* |
| DistAug (Jun et al., 2020) | 2.53 | - |
| Very Deep VAE (Child, 2021) | 2.87 | 3.80* |
| PixelSNAIL (Chen et al., 2018) | 2.85 | 3.80* |
| ScoreFlow (Song et al., 2021) | 2.83 | 3.76* |
| VDM (Kingma et al., 2021) | 2.65 | 3.72* |
| VDM (w/ uniform dequantization) | 2.85 | 3.76 |
| VDM (w/ data augmentation) | 2.49 | - |
| i-DODE (Zheng et al., 2023) | 2.56 | 3.43 |
| i-DODE (w/ data augmentation) | 2.42 | - |
| MULAN (Sahoo et al., 2024) | 2.55 | 3.67 |
| **QDPM (ours)** | **0.27** | **0.32** |
| Theoretical lower bound | 0.0043 | 0.0051 |

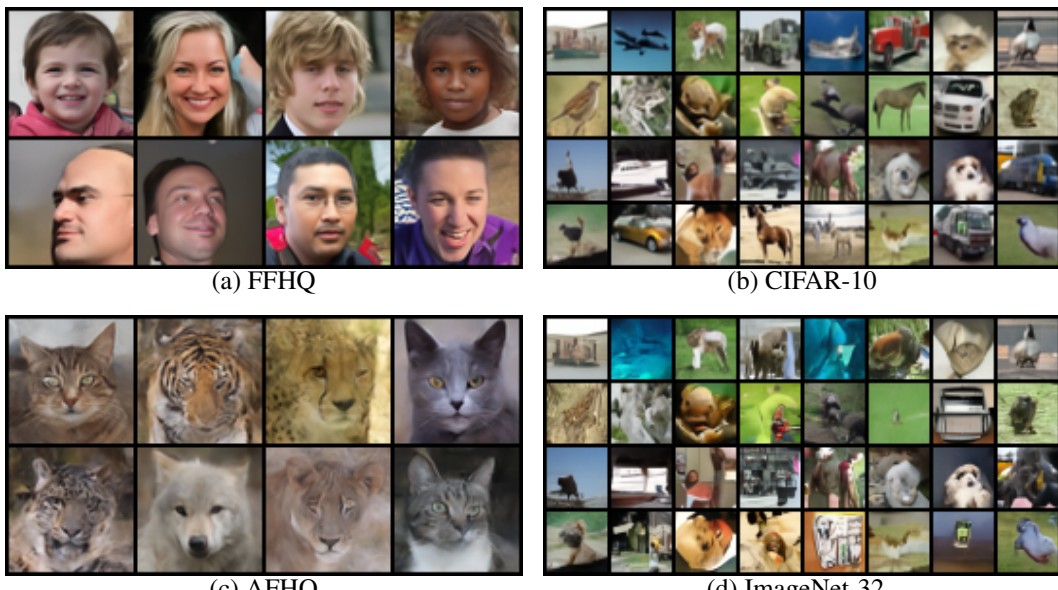

(a) FFHQ      (b) CIFAR-10

(c) AFHQ      (d) ImageNet-32

Figure 4: Non-cherrypicked samples of QDPM for (a) Flickr-Faces-HQ (FFHQ), (b) CIFAR-10, (c) Animal-Faces-HQ (AFHQ), and (d) ImageNet-32 dataset.

sponds to the NLL scaled by the data dimension $d$ with the log base 2 as follows:

$$\text{BPD} = -\frac{1}{d} \mathbb{E}_{p_{\text{data}}} \left[ \log_2 p\left(\mathbf{x}; \theta\right) \right] = -\frac{1}{d \log 2} \mathbb{E}_{p_{\text{data}}} \left[ \log p\left(\mathbf{x}; \theta\right) \right]. \tag{33}$$

For our QDPM, we calculate the upper bound of the BPD using Eq. (24). We also report the Fréchet inception distance (FID) of the generated images via SDE solver for the evaluation of perceptual quality, although we do not tune our QDPM for it, since our main focus is on density estimation. Since the negative log-likelihood for the reverse SDE is intractable, we report its upper bound calculated with Eq. (19).

The results are summarized in Table 5. Our QDPM demonstrates the state-of-the-art result for both CIFAR-10 and ImageNet-32 dataset in terms of the test likelihood. The test negative log-likelihood (NLL) scores are improved from the previous SoTA methods over 2.0 (bpd). Since CIFAR-10 has 10,000 test images with the resolution of $32 \times 32$ of RGB color channels, the theoretical lower bound of the test NLL is $\log_2\left(10000\right) / \left(3 \times 32^2\right) \approx 0.0043$ (see Appendix C for the derivation). To the

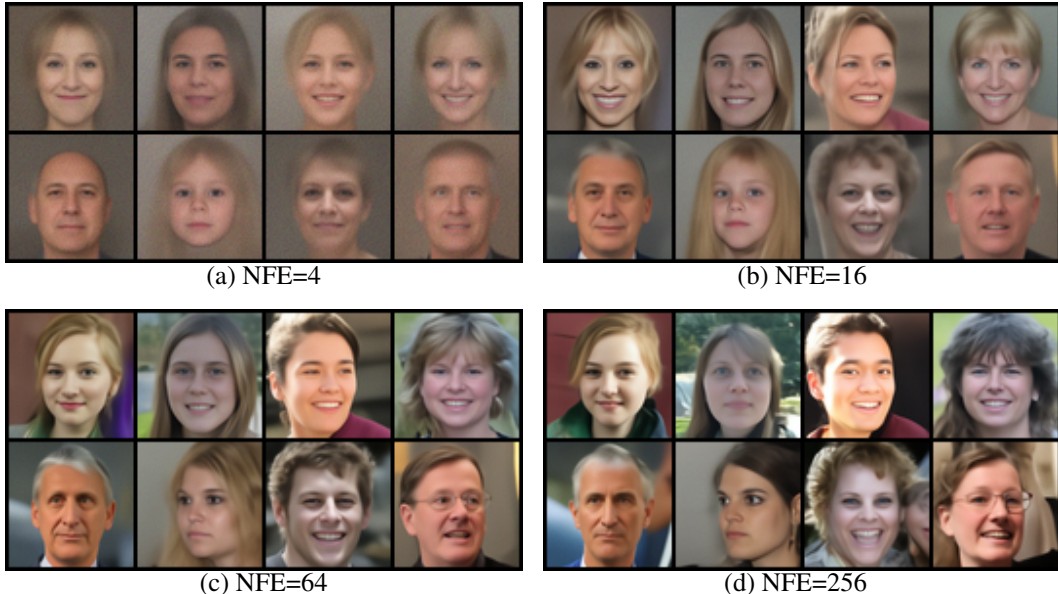

(a) NFE=4                                  (b) NFE=16

(c) NFE=64                                 (d) NFE=256

Figure 5: FFHQ samples by QDPM-Solver-2 with number of function evaluations (NFEs) in a range of $4 \sim 256$.

best of our knowledge, our QDPM is the first result that achieves the test NLL under $1.0$, approaching the theoretical lower bound. For qualitative evaluation, generated samples by our QDPMs are shown in Figure 4. It can be seen that our QDPM can generate high-quality samples without cherrypicking. Regarding the FID score, we observe $5.60$ for CIFAR-10 and $8.89$ for ImageNet-32. These are still worse than the SoTA FID scores, such as $1.54$ of GDD (Zheng & Yang, 2024) for CIFAR-10. Previous works also have reported that the test NLL and FID score do not tend to correlate, and diffusion models trained with a maximum likelihood objective tend to show worse FID scores (Ho et al., 2020; Song et al., 2021; Zheng et al., 2023); hence, this is a natural phenomenon. We also provide a qualitative comparison of generated samples when the number of function evaluations (NFEs) changes in Figure 5. We use the second-order QDPM-Solver, so NFE corresponds to twice the number of discretization steps (i.e., $2 \times M$ in Algorithm 2). It can be observed that our QDPM-Solver can generate high-quality samples even with a relatively small NFEs (e.g., $16 \sim 64$).

## 6  CONCLUSION

In this paper, we propose the Quantizing Diffusion Probabilistic Model (QDPM), whose generated samples are guaranteed to converge to a quantized point. This is achieved by designing the signal prediction model so that its fixed points corresponds to quantized points as described in Section 3. We also provide efficient SDE solvers for QDPM, which are derived using the closed-form solution of QDPM's reverse SDE. We experimentally show that our QDPM can substantially improve the performance of diffusion model for image density estimation, and achieves the state-of-the-art results for test negative log-likelihood by a large margin. While we have focused on the SDE-based formulation throughout the paper, we believe it can be extended to an ODE-based formulation as in flow matching; we leave this for future work. In addition, developing more efficient solvers with higher-order convergence guarantee is also a promising direction for further improvement.

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

## A  PROOFS

**Proposition 3** (Formal restatement of Proposition 1). *Let $\mathbf{x}_t$ be a solution on $t \in [0, \infty)$ of the SDE*

$$d\mathbf{x}_t = \frac{2}{t}\left(\mathbf{x}_t - \hat{\mathbf{x}}_\theta\left(\mathbf{x}_t, t\right)\right) dt + \sqrt{2t}d\mathbf{w}_t, \tag{34}$$

*and assume:*

1. $\mathbf{x}_t$ *admits a (random) continuous extension at $t = 0$: $\mathbf{x}_t \to \mathbf{x}_0$ as $t \downarrow 0$ almost surely.*

2. $\hat{\mathbf{x}}_\theta : \mathbb{R}^d \times [0, \infty) \to \mathbb{R}^d$ *is continuous at $(\mathbf{x}_0, 0)$ and locally bounded in a neighborhood of $(\mathbf{x}_0, 0)$.*

*Then, the following holds almost surely:*

$$\mathbf{x}_0 = \hat{\mathbf{x}}_\theta\left(\mathbf{x}_0, 0\right). \tag{35}$$

*Proof.* Fix $0 \leq t < s < \infty$. Let $g(t) = t^{-2}$ (deterministic, $C^1$ on $[0, \infty)$) and apply the integration-by-parts for semimartingales:

$$d\left(g(t)\mathbf{x}_t\right) = g(t)d\mathbf{x}_t + \mathbf{x}_t dg(t) + d[g, \mathbf{x}]_t. \tag{36}$$

Since $g$ has finite variation, its quadratic covariation with any continuous semimartingale is zero, i.e. $[g, x] \equiv 0$. Using $dg(t) = -2t^{-3}dt$ and the given SDE,

$$d\left(\frac{\mathbf{x}_t}{t^2}\right) = -\frac{2}{t^3}\hat{\mathbf{x}}_\theta\left(\mathbf{x}_t, t\right) dt + \sqrt{2}t^{-3/2}d\mathbf{w}_t. \tag{37}$$

We used the semimartingale product rule and the facts that finite-variation processes have zero quadratic variation and that $[g, x] = 0$. Integrating from $s$ to $t$ yields,

$$\frac{\mathbf{x}_t}{t^2} - \frac{\mathbf{x}_s}{s^2} = -\int_s^t \frac{2}{r^3}\hat{\mathbf{x}}_\theta\left(\mathbf{x}_r, r\right) dr \int_s^t \sqrt{2}r^{-3/2}d\mathbf{w}_r. \tag{38}$$

Multiplying by $s^2$ and rearrange:

$$\mathbf{x}_t = \frac{t^2}{s^2}\mathbf{x}_s - t^2\int_s^t \frac{2}{r^3}\hat{\mathbf{x}}_\theta\left(\mathbf{x}_r, r\right) dr + t^2\int_s^t \sqrt{\frac{2}{r^3}}d\mathbf{w}_r. \tag{39}$$

We now let $t \downarrow 0$ with $s > 0$ fixed and evaluate each term:

- For the first term, $\dfrac{t^2}{s^2}\mathbf{x}_s \to 0$ almost surely.

- For the stochastic term, by Itô isometry,

$$\mathbb{E}\left[\left(t^2\int_t^s \sqrt{2}r^{-3/2}d\mathbf{w}_r\right)^2\right] = t^4\int_t^s 2r^{-3}dr = t^2\left(1 - \frac{t^2}{s^2}\right) \xrightarrow[t\downarrow 0]{} 0. \tag{40}$$

  so this term converges to 0 in $L^2$ (hence in probability, along a subsequence almost surely).

- For the deterministic integral term, set $h(r) := \hat{\mathbf{x}}_\theta(\mathbf{x}_r, r)$. By the second assumption and $\mathbf{x}_r \to \mathbf{x}_0$, we have $h(r) \to \hat{\mathbf{x}}_\theta(\mathbf{x}_0, 0)$ as $r \downarrow 0$, and $h$ is locally bounded near 0. For any small $\delta \in (0, s]$,

$$t^2\int_t^s \frac{2}{r^3}h(r)dr = 2t^2\int_t^\delta \frac{h(r)}{r^3}dr + 2t^2\int_\delta^s \frac{h(r)}{r^3}dr. \tag{41}$$

  The second integral tends to 0 since it is bounded by $2Ct^2\int_\delta^s r^{-3}dr$. For the first, write $h(r) = h(0) + \varepsilon(r)$ with $\varepsilon(r) \to 0$ as $r \downarrow 0$. Then

$$2t^2\int_t^\delta \frac{h(0)}{r^3}dr + 2t^2\int_t^\delta \frac{\varepsilon(r)}{r^3}dr = h(0)\left(1 - \frac{t^2}{\delta^2}\right)2t^2\int_t^\delta \frac{\varepsilon(r)}{r^3}dr. \tag{42}$$

  Given $\eta > 0$, choose $\delta$ so small that $\sup_{0 < r \leq \delta}|\varepsilon(r)| < \eta$; then the last term is bounded by $\eta\left(1 - \frac{t^2}{\delta^2}\right)$, which can be made $< \eta$ uniformly in $t$. Hence the whole expression tends to $h(0) = \hat{\mathbf{x}}_\theta(\mathbf{x}_0, 0)$ as $t \downarrow 0$.

Taking limits in Eq. (39) and using $\mathbf{x}_t \to \mathbf{x}_0$ by the first assumption yields $\mathbf{x}_0 = \hat{\mathbf{x}}_\theta(\mathbf{x}_0, 0)$ almost surely. $\qquad\qquad\square$

**Proposition 4** (Formal restatement of Proposition 2). *Let* $\Omega = \left( \{x^{(k)}\}_{k=1}^K \right)^d \subset \mathbb{R}^d$, *be fixed.*
*Define a rounding map* $\mathrm{round} : \mathbb{R}^d \to \Omega$ *by* $\mathrm{round}(x) \in \arg\min_{y \in \Omega} \|x - y\|_2$, *with a deterministic tie–breaking rule so that* $\mathrm{round}$ *is single-valued (equivalently, a single-valued selection of the metric projection onto the finite set* $\Omega$*). Under the same assumptions with Proposition 3, if for all* $\mathbf{x} \in \mathbb{R}^d$*,* $\hat{\mathbf{x}}_\theta(\mathbf{x}, 0) = \mathrm{round}(x)$*, the limit* $\mathbf{x}_0$ *takes values in the codebook, i.e.,* $\mathbf{x}_0 \in \Omega$ *almost surely.*

*Proof.* By Proposition 3, any such solution satisfies $\mathbf{x}_0 = \hat{\mathbf{x}}_\theta(\mathbf{x}_0, 0)$ almost surely. Invoking the fact $\hat{\mathbf{x}}_\theta(\mathbf{x}, 0) = \mathrm{round}(x)$, the following holds:

$$\mathbf{x}_0 = \mathrm{round}(\mathbf{x}_0) \in \Omega \quad \text{a.s.} \tag{43}$$

Finally, because $\mathrm{round}$ maps $\mathbb{R}^d$ into $\Omega$, the equality $\mathrm{round}(\mathbf{x}) = \mathbf{x}$ can hold only when $\mathbf{x} \in \Omega$: if $\mathbf{x} \notin \Omega$, then $\mathrm{round}(\mathbf{x}) \in \Omega \neq \mathbf{x}$. Hence $\mathbf{x}_0 \in \Omega$ almost surely. $\qquad\square$

## B    RELATION TO EQILIBRIUM MATCHING

Recently, Wang & Du (2025) have proposed the equilibrium matching model (EqM), whose loss function is defined as follows:

$$\mathcal{L}_{\mathrm{EqM}} = \mathbb{E}\left[ \|f_\theta(\tilde{\mathbf{x}}_\gamma) - (\boldsymbol{\epsilon} - \mathbf{x}_0) \cdot c(\gamma)\|^2 \right], \tag{44}$$

where $\gamma \sim \mathcal{U}(0, 1)$ and $\tilde{\mathbf{x}}_\gamma = \gamma \mathbf{x}_0 + (1 - \gamma) \boldsymbol{\epsilon}$. By defining $\hat{\mathbf{x}}_\theta(\tilde{\mathbf{x}}_\gamma, \gamma) = \tilde{\mathbf{x}}_\gamma - \frac{1-\gamma}{c(\gamma)} f_\theta(\tilde{\mathbf{x}}_\gamma)$, this can be transformed into the form of the signal prediction loss as follows:

$$\mathcal{L}_{\mathrm{EqM}} = \mathbb{E}\left[ \left\| f_\theta(\tilde{\mathbf{x}}_\gamma) - \left( \frac{1}{1-\gamma}(\tilde{\mathbf{x}}_\gamma - \gamma \mathbf{x}_0) - \mathbf{x}_0 \right) \cdot c(\gamma) \right\|^2 \right] \tag{45}$$

$$= \mathbb{E}\left[ \left\| f_\theta(\tilde{\mathbf{x}}_\gamma) - \frac{c(\gamma)}{1-\gamma}(\tilde{\mathbf{x}}_\gamma - \mathbf{x}_0) \right\|^2 \right] \tag{46}$$

$$= \mathbb{E}\left[ \left\| \frac{c(\gamma)}{1-\gamma}\left( \mathbf{x}_0 - \left( \tilde{\mathbf{x}}_\gamma - \frac{1-\gamma}{c(\gamma)} f_\theta(\tilde{\mathbf{x}}_\gamma) \right) \right) \right\|^2 \right] \tag{47}$$

$$= \mathbb{E}\left[ \frac{c(\gamma)^2}{(1-\gamma)^2} \|\mathbf{x}_0 - \hat{\mathbf{x}}_\theta(\tilde{\mathbf{x}}_\gamma, \gamma)\|^2 \right] \tag{48}$$

In addition, by defining $t = (1 - \gamma)/\gamma$, $\mathbf{x}_t = \tilde{\mathbf{x}}_\gamma / \gamma$, and $\hat{\mathbf{x}}_\theta(\mathbf{x}_t, t) := \hat{\mathbf{x}}_\theta(\tilde{\mathbf{x}}_\gamma = \mathbf{x}_t/(1 + t), \gamma = 1/(1 + t))$ with a little abuse of notation, we can recover a similar form to Eq. (19):

$$\mathcal{L}_{\mathrm{EqM}} = \int_0^1 \frac{c(\gamma)^2}{(1-\gamma)^2} \|\mathbf{x}_0 - \hat{\mathbf{x}}_\theta(\tilde{\mathbf{x}}_\gamma, \gamma)\|^2 \, d\gamma \tag{49}$$

$$= \int_0^\infty \frac{c(1/(1+t))^2}{t^2} \|\mathbf{x}_0 - \hat{\mathbf{x}}_\theta(\mathbf{x}_t, t)\|^2 \, dt. \tag{50}$$

When $c(\gamma) = \sqrt{\gamma/(1 - \gamma)}$, this is exactly the same with Eq. (19). Therefore, the difference between EqM and our QDPM can be absorbed into the design choices of the signal predictor $\hat{\mathbf{x}}_\theta$ and the weighting function $c$.

## C    DERIVATION OF THEORETICAL LOWER BOUND OF NLL

First, we show that the log-likelihood of the model distribution $p(\mathbf{x})$ for a finite set of data $\mathbf{x}[1], \ldots, \mathbf{x}[N]$ is maximized when it corresponds to the empirical distribution of $\mathbf{x}[1], \ldots, \mathbf{x}[N]$.

**Proposition 5.** *Let $\mathcal{X} = \{a_1, \ldots, a_M\}$ be a finite set, and let*

$$\mathbf{x}[1], \ldots, \mathbf{x}[N] \in \mathcal{X} \tag{51}$$

*be observed data points. For each $m \in \{1, \ldots, M\}$, define the empirical frequency*

$$n_m := \#\{i \in \{1, \ldots, N\} : \mathbf{x}[i] = a_m\}, \quad \hat{p}(a_m) := \frac{n_m}{N}. \tag{52}$$

*Thus $\hat{p}$ is the empirical distribution of the sample. Consider any probability mass function $p$ on $\mathcal{X}$ satisfying*

$$p(a_m) > 0 \quad \text{whenever } n_m > 0. \tag{53}$$

*Define the empirical average log-likelihood of $p$ as*

$$L(p) := \frac{1}{N} \sum_{i=1}^{N} \log p(\mathbf{x}[i]). \tag{54}$$

*Then $L(p)$ is maximized over all such $p$ uniquely at $p = \hat{p}$; that is,*

$$L(p) \leq L(\hat{p}) \quad \text{for all admissible } p, \tag{55}$$

*with equality if and only if $p = \hat{p}$.*

*Proof.* First rewrite $L(p)$ in terms of the empirical distribution. Using the counts $n_m$,

$$L(p) = \frac{1}{N} \sum_{i=1}^{N} \log p(\mathbf{x}[i]) = \frac{1}{N} \sum_{m=1}^{M} n_m \log p(a_m) = \sum_{m=1}^{M} \hat{p}(a_m) \log p(a_m). \tag{56}$$

Now consider the Kullback–Leibler divergence from $\hat{p}$ to $p$:

$$D_{\mathrm{KL}}(\hat{p} \,\|\, p) := \sum_{m=1}^{M} \hat{p}(a_m) \log \frac{\hat{p}(a_m)}{p(a_m)}. \tag{57}$$

This can be expanded as

$$D_{\mathrm{KL}}(\hat{p} \,\|\, p) = \sum_{m=1}^{M} \hat{p}(a_m) \log \hat{p}(a_m) - \sum_{m=1}^{M} \hat{p}(a_m) \log p(a_m). \tag{58}$$

Rearranging gives

$$\sum_{m=1}^{M} \hat{p}(a_m) \log p(a_m) = \sum_{m=1}^{M} \hat{p}(a_m) \log \hat{p}(a_m) - D_{\mathrm{KL}}(\hat{p} \,\|\, p). \tag{59}$$

The left-hand side is precisely $L(p)$, so

$$L(p) = \sum_{m=1}^{M} \hat{p}(a_m) \log \hat{p}(a_m) - D_{\mathrm{KL}}(\hat{p} \,\|\, p). \tag{60}$$

The first term $\sum_m \hat{p}(a_m) \log \hat{p}(a_m)$ depends only on the empirical distribution $\hat{p}$ and not on $p$; hence, it is a constant with respect to $p$.

On the other hand, by the non-negativity of KL divergence,

$$D_{\mathrm{KL}}(\hat{p} \,\|\, p) \geq 0, \tag{61}$$

with equality if and only if $p = \hat{p}$. Therefore,

$$L(p) = (\text{constant}) - D_{\mathrm{KL}}(\hat{p} \,\|\, p) \leq (\text{constant}) - 0 = L(\hat{p}), \tag{62}$$

and equality holds if and only if $D_{\mathrm{KL}}(\hat{p} \,\|\, p) = 0$, i.e., $p = \hat{p}$. Thus, $L(p)$ is maximized uniquely at the empirical distribution $p = \hat{p}$. $\qquad \square$

Since CIFAR-10 have 10,000 test images, the likelihood of the test set is maximized when the following holds:

$$p^* (\mathbf{x}; \theta) = \begin{cases} \frac{1}{10000} & \text{if} \quad \mathbf{x} \in \mathbb{X}_{\text{test}} \\ 0 & \text{otherwise,} \end{cases} \tag{63}$$

where $\mathbb{X}_{\text{test}}$ is a set of test examples with $|\mathbb{X}_{\text{test}}| = 10000$. Note that we assume that there is no overlap inside the dataset, which is confirmed in a prior work for CIFAR-10 (Barz & Denzler, 2020)[2]. Therefore, the theoretical lower bound of the test NLL as follows:

$$\text{NLL}^* = -\frac{1}{|\mathbb{X}_{\text{test}}|} \sum_{\mathbf{x} \in \mathbb{X}_{\text{test}}} \log p^* (\mathbf{x}; \theta) \tag{64}$$

$$= -\log 10000. \tag{65}$$

By transforming it to the scale of bits per dimension (BPD), we can derive the theoretical lower bound of NLL in the BPD scale as follows:

$$\text{BPD}^* = \frac{1}{|\mathbb{X}_{\text{test}}| \cdot \dim(\mathbf{x})} \sum_{\mathbf{x} \in \mathbb{X}_{\text{test}}} \log_2 p^* (\mathbf{x}; \theta) \tag{66}$$

$$= \frac{1}{\dim(\mathbf{x}) \cdot \log 2} \text{NLL}^* \tag{67}$$

$$= \frac{1}{3 \times 32 \times 32} \log_2 10000 \tag{68}$$

$$\approx 0.0043. \tag{69}$$

Similarly, the NLL lower bound for ImageNet-32, which has 50,000 test images, can be derived as follows:

$$\text{BPD}^* = \frac{1}{3 \times 32 \times 32} \log_2 50000 \tag{70}$$

$$\approx 0.0051. \tag{71}$$

# D DETAILS OF EXPERIMENTAL SETUPS

## D.1 CODE

Our implementation for the experiment is available at `https://anonymous.4open.science/r/qdpm-iclr2026-D50E`.

## D.2 TOTAL AMOUNT OF COMPUTE

We run our experiments mainly on cloud GPU instances with $8\times$ H100. It took approximately 120 hours for our experiments in total.

## D.3 LICENSE OF ASSETS

**Datasets:** The terms of access for the CIFAR-10 database is provided at `https://www.cs.toronto.edu/~kriz/cifar.html` The terms of access for the ImageNet database is provided at `https://www.image-net.org/download`. The terms of access for the Flickr-Faces-HQ (FFHQ) database is provided at `https://github.com/NVlabs/ffhq-dataset/blob/master/LICENSE.txt`. The terms of access for the Animal-Faces-HQ (AFHQ) database is provided at `https://github.com/clovaai/stargan-v2/blob/master/LICENSE`

**Code:** Our implementation is based on the official Jax code of Kingma et al. (2021) provided at `https://github.com/google-research/vdm`. We reimplement it with PyTorch, and modify for our experimental settings.

---

[2]For ImageNet, there is no clear evidense that there is no overlap to the best of our knowledge, so the BPD lower bound can be a little smaller than 0.0051.

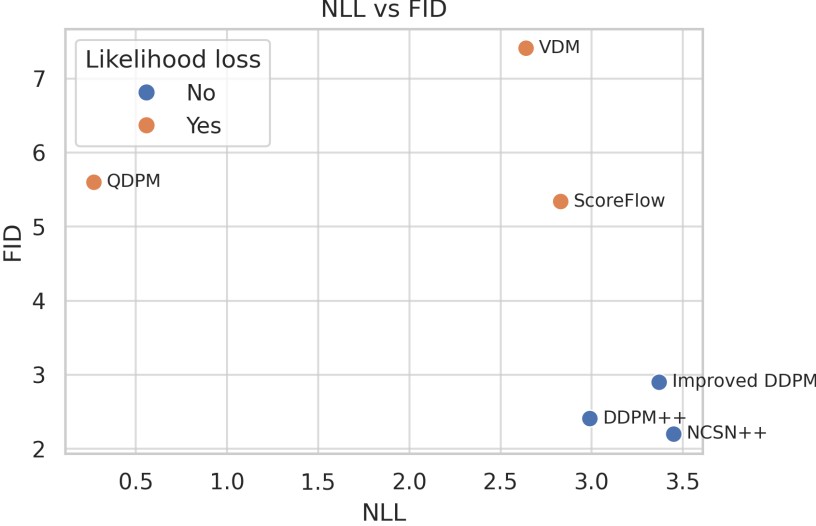

Figure 6: Trade-off between negative log-likelihood (NLL) and Frechet Inception Distance (FID).

# E  THE USE OF LARGE LANGUAGE MODELS

We used a large language model (ChatGPT; "GPT-5 Thinking") only for English-language polishing, i.e., grammar, phrasing, and style improvements, and light copy-editing (e.g., reducing redundancy, harmonizing terminology). The model was not used to generate ideas, derivations, proofs, experimental designs, results, figures, tables, or code, nor to select or fabricate references. All technical content and claims were authored and verified by the human authors, who edited every LLM suggestion and take full responsibility for the final text. No non-public data, confidential information, or personally identifiable information were provided to the model beyond portions of the manuscript text necessary for editing.

