# OpenReview forum: "Quantization-Aware Diffusion Models For Maximum Likelihood Training"
_ICLR.cc/2026/Conference — ICLR 2026 Poster_

### Official Review · Reviewer_12ST · 2025-11-01

**Soundness:** 3
**Presentation:** 3
**Contribution:** 3
**Rating:** 6
**Confidence:** 2

**Summary:**

This work introduces a new parameterization technique for diffusion models aimed at improving negative log-likelihood performance by ensuring generated outputs lie on quantized values. The approach is broadly applicable to density modeling on discretized data and experimental results show significant NLL gains.

**Strengths:**

1. The paper is clearly written with well-motivated methodology. The rationale behind the approach and its implementation is easy to understand, and the experiments effectively highlight its advantages.

2. The method is supported by theoretical analysis regarding both the parameterization and associated solvers, though there may be some concerns with the proposition as noted below.

3. Strong experimental results are demonstrated on discrete-signal density estimation tasks, outperforming prior state-of-the-art methods. Notably, CIFAR-10 results show substantial improvement.

**Weaknesses:**

A recent work [1] proposes an equilibrium formulation for image generation rather than traditional density estimation. Although their approach does not strictly ensure convergence to quantized points, it explicitly defines a fixed point for the generative process. It would be valuable to discuss how this relates to your method and whether their training strategy could potentially benefit or be adapted to your framework. In addition, that work argues that standard diffusion models do not exhibit a valid fixed point at t=0. This observation seems relevant to Proposition 1 in your paper and merits further discussion or clarification.

2. In the proof, the function labeled $f$ should be $\hat{x}_\theta$

[1] Wang, Runqian, and Yilun Du. "Equilibrium Matching: Generative Modeling with Implicit Energy-Based Models." arXiv preprint arXiv:2510.02300 (2025).

**Questions:**

NA

---

> ### Author Response · Authors · 2025-11-21
> **Response to Reviewer 12ST**
>
> Thank you for your insightful feedback.
>
> > A recent work [1] proposes an equilibrium formulation for image generation rather than traditional density estimation. Although their approach does not strictly ensure convergence to quantized points, it explicitly defines a fixed point for the generative process. It would be valuable to discuss how this relates to your method and whether their training strategy could potentially benefit or be adapted to your framework. In addition, that work argues that standard diffusion models do not exhibit a valid fixed point at t=0. This observation seems relevant to Proposition 1 in your paper and merits further discussion or clarification.
>
> Thank you for your insightful suggestion. We have added a discussion on the relation to the equilibrium matching model (EqM) in Appendix B. In a nutshell, the relation between EqM and our QDPM can be explained by the difference of their definitions for the signal predictor and the time-dependent weighting of the loss function. Please refer to the updated paper for more details. In terms of discussion on the fixed point, it seems that they use a slightly different definition of the fixed point with ours. In our paper, we discuss the fixed point of the signal predictor, at which $\hat{x} ( x ) = x$ holds. On the other hand, they discuss the fixed point of the energy landscape, at which $\nabla_x E ( x ) = - \hat{s} ( x , 0) = 0$. Investigating the relation between these two can be an interesting direction of future work.
>
> > In the proof, the function labeled  should be
>
> Thank you for pointing it out. We have fixed these typos in the updated version.
>
>
> Feel free to ask further questions if you may have.
>
> Thanks.

---

> > ### Comment · Reviewer_12ST · 2025-11-27
> >
> > Thanks for the explanation, I will keep my current rating.

---

> > > ### Author Response · Authors · 2025-11-28
> > >
> > > Thank you for your follow-up comments.
> > >
> > > Feel free to ask further questions if you may have.
> > >
> > > Thanks.

---

### Official Review · Reviewer_Zbvq · 2025-11-09

**Soundness:** 3
**Presentation:** 3
**Contribution:** 3
**Rating:** 6
**Confidence:** 2

**Summary:**

This paper introduces a new parameterization for diffusion models designed to improve negative log-likelihood (NLL) performance by ensuring that generated samples converge to quantized points. The proposed parameterization can be broadly applied to density estimation tasks involving quantized data. Experimental results demonstrate substantial improvements in NLL across various datasets.

**Strengths:**

* The paper is clearly written and easy to follow. The motivation and methodological design are well articulated, and the experiments convincingly demonstrate the proposed approach’s effectiveness.

* The method is supported by theoretical analysis for both the parameterization and the associated solvers, though there are some issues with the description in Proposition 1 (see Weaknesses).

* The experiments show strong performance on density estimation for discrete signal data, outperforming state-of-the-art methods. For example, the improvement on CIFAR-10 reduces NLL from 2.42 to 0.27.

**Weaknesses:**

* Proposition 1 claims that the SDE defined in Eq. (14) has a fixed point at time step 0; however, this does not appear to hold in the general case. Constructing denoised score matching for diffusion models requires defining the score field over discrete data through noise injection. As discussed in [1], when the model distribution approaches the real data distribution, the gradient becomes steeper due to the small Gaussian variance, making the boundedness assumption not always valid. While the proposed parameterization may alleviate this issue, a more detailed explanation covering broader cases would strengthen the argument. Moreover, the role of the fixed-point requirement in Proposition 1 is unclear.

* Beyond the explanation in Proposition 1, an ablation study validating this proposition, for instance, by reporting the magnitude of the fixed-point error, would significantly improve the paper’s empirical evidence.
​

[1] Song, Yang, et al. "Sliced Score Matching: A Scalable Approach to Density and Score Estimation." Uncertainty in Artificial Intelligence. PMLR, 2020.

**Questions:**

* In the proof, should $f$ be $\hat{x}_{\theta}$?

* Could you provide more details about the role of the fixed-point requirement in Proposition 1?

---

> ### Author Response · Authors · 2025-11-21
> **Response to Reviewer Zbvq**
>
> Thank you for giving us insightful feedback. Below is our response to address your concerns.
>
> > Proposition 1 claims that the SDE defined in Eq. (14) has a fixed point at time step 0; however, this does not appear to hold in the general case. Constructing denoised score matching for diffusion models requires defining the score field over discrete data through noise injection. As discussed in [1], when the model distribution approaches the real data distribution, the gradient becomes steeper due to the small Gaussian variance, making the boundedness assumption not always valid. While the proposed parameterization may alleviate this issue, a more detailed explanation covering broader cases would strengthen the argument. Moreover, the role of the fixed-point requirement in Proposition 1 is unclear.
>
>
>
> As you pointed out, when training diffusion models with a finite training set of discrete data, the gradient (i.e., score $s_\theta$ ) tends to become very steep around $t = 0$.
> However, that is not the case for the signal predictor $ \hat{x} $, which is an estimator of $\mathbb{E} [ x_0 \mid x_t ]$. This can be also explained from the definition $\hat{x} = x_t - t \cdot \hat{\epsilon} = x_t + t^2 \cdot \hat{s}$.
> Even when the score $\hat{s}$ diverges, the signal predictor does not diverge since $\hat{s}$ does not affect $\hat{x}$ around $t = 0$.
>
> In terms of the role of the fixed-point requirement in Proposition 1, it is critical to construct our QDPM, because it can be used to ensure that the reverse SDE converges to a quantized point if $\hat{x} ( x , 0 ) \in \Omega$ holds for any $x$ as stated in Proposition 2.
>
> > Beyond the explanation in Proposition 1, an ablation study validating this proposition, for instance, by reporting the magnitude of the fixed-point error, would significantly improve the paper’s empirical evidence. ​
>
> Thank you for your suggestion. To verify whether Proposition 1 holds in practice, we measure the fixed-point error at each time step when sampling from the pretrained EDM on CIFAR-10 and AFHQ and plot it in Figure 3. As can be seen from it, the fixed-point error converges to zero as $t \to 0$.
>
> Feel free to ask further questions if you may have.
>
> Thanks.

---

> ### Author Response · Authors · 2025-11-28
> **Gentle Reminder to Reviewer Zbvq**
>
> Thank you again to your efforts in reviewing our paper and your constructive comments. This is a gentle reminder that we have updated the paper to address your concerns. The discussion period will end in five days, so please let us know if you have further comments about the update.
>
> Thanks!

---

### Official Review · Reviewer_Pzwu · 2025-11-09

**Soundness:** 3
**Presentation:** 3
**Contribution:** 3
**Rating:** 6
**Confidence:** 4

**Summary:**

The work introduces a quantization-aware diffusion parameterization that guarantees convergence to discrete codebook points at
$t \downarrow 0$. It derives a reverse-SDE solution specific to this parameterization, proposes first/second-order solvers, and trains with a likelihood-motivated objective that avoids explicit dequantization. Results show very low bits-per-dim (BPD) on CIFAR-10 and ImageNet-32.

In more detail, the paper proposes a parameterization that forces the reverse SDE to converge to quantized points by making $x_\theta(\cdot,0)$ equal a (soft) rounding operator; formally, the limit $x_0$ is a fixed point of $x_\theta(\cdot,0)$, and they limit the set of fixed points to a finite set, hence the output is quantized. The training objective is presented as an ELBO-style upper bound on $-\log p_0(x_0)$ with $t_{\min} \to 0$, $t_{\max} \to\infty$ (Eq. 19), then reparameterized via $u=e^{-1/t^2}$ to yield a simple Monte-Carlo form (Eq. 22). They evaluate an upper bound on NLL for the reverse SDE and report extremely low BPD on CIFAR-10/ImageNet-32 (0.27 / 0.32), claiming closeness to a “theoretical lower bound”.

**Strengths:**

- Clear mechanism to force quantized fixed points; the softround + $e^{-1/t^2}\delta_\theta$ parameterization is simple and elegant.
- Derivation of an SDE solution enables specialized solvers with straightforward implementations.
- Ambitious likelihood-centric evaluation; if comparable, the reported BPD would be a genuine leap.

**Weaknesses:**

None listed yet; see **Questions** instead. I prefer to label weaknesses only after the discussion period. Everything is only a question until then.

**Questions:**

### 0. A few typos

Throughout, there are expressions of the form $\Omega = \prod_{i=1}^d \{ x^{(k)}\}_{k=1}^K$ where $i$ is not used

### 1. Comparability of the “likelihood” metric

Prior continuous-model papers typically report *dequantized* likelihoods (e.g., uniform $[0,1)$ or learned dequantization), or use a categorical $p_\theta(x_0 \mid x_{t_{\min}})$ at a positive $t_{\min}$. Here the bound integrates to $t_{\min} \to 0$ with a quantized fixed-point parameterization.  It isn’t obvious this produces the same quantity other works report.  It would help to clarify whether the contribution is primarily the bound, the trained model, or both.

**(1a) Type of bound**

Please state at a high level whether your BPD corresponds to:

(i) discrete likelihood (categorical over 256 bins per channel),
(ii) dequantized continuous likelihood, or


**(1b) Logp → BPD conversion**

Right now the text says you “report [the] upper bound calculated with Eq. (19)” but does not specify the unit conversion or definition used to go from log-likelihood to BPD.  Assuming the logp is computed correctly, could you specify the exact computation you used to convert from your logp bound to your BPD bound?

**(1c) Evaluating other models with your bound, and evaluating your model with other bounds**

- (1c.i) Would your bound, if used to evaluate another model, also report numbers <=1.0? Especially given that rounding is included in the likelihood evaluation. Regardless of how another model checkpoint was trained, does this bound (with rounding) give that model a better BPD too? Could this be evaluated on existing checkpoints from Song, Kingma, etc.?

- (1c.ii) Conversely, what would the *standard* dequantization bound (Eq. 26) report for your model, without the rounding step?

### 2. (Important!) Change-of-variables detail in Eq. (22)

Question 1 concerned defining the BPD (assuming logp computation is correct).  This question is about the correctness of the logp computation itself. When moving from Eq. (20) to Eq. (22) with the substitution $u=e^{-1/t^2}$, the weighting induced by $du/dt$ must be accounted for; otherwise the Monte-Carlo estimator could be biased and underestimate NLL.   Could you please show the exact steps (including Jacobian factors) to justify the *unweighted* expectation over $u \sim \mathrm{Unif}(0,1)$ in Eq. (22)?

### 3. “Theoretical lower bound” – Q1

The lower bound $\log_2(50000)/(3 \cdot 32^2) \approx 0.0051$ appears to be the code-length of a uniform categorical over the 50k images divided by dimension.  This is a property of the finite dataset, not the underlying data distribution. Please justify using this as a target for generative modeling and clarify whether your evaluation implicitly allows near-memorization (e.g., concentrating mass on quantized locations). Is it a theoretical lower bound on the 50k images?

### 4. “Theoretical lower bound” – Q2

Why is the word “test” used here? For CIFAR there are 50k train images and 10k test images; for ImageNet, ~1.2M train and 50k test.
Was there a mix-up? The other numbers reported in Table 1 usually refer to the CIFAR *test* set of 10,000 datapoints (even though FID is often reported on the training set). Which sets were your BPD numbers computed on?  Why do we compare CIFAR test BPDs to a lower bound computed on 50k training points?

### 5. Definition of $s_t$

Could you explicitly define $s_t(x_t) = \nabla_{x_t} \log q_{0t}(x_t \mid x_0)$ around Eqs. (2)–(4) for readability? It may be clear to experts, but all mathematical symbols should be defined.

### 6. Code availability

The anonymous code link in Appx. B.1 is currently empty.  It would be helpful to share working code so others can confirm the BPD computation protocol.

### 7. Sensitivity of reported BPD

How sensitive are the reported BPD values to the time-discretization schedule? In my experience, sampling 1,000 vs. 10,000 uniform points in (0,1) can drastically affect dequantized-diffusion BPDs for standard noise schedules and common checkpoints (e.g., those from Song, Durkan & Song, Kingma, etc.).

---

> ### Author Response · Authors · 2025-11-21
> **Response to Reviewer Pzwu (Part 1)**
>
> We thank the reviewer for taking your time and giving insightful feedback.
>
> > 0. A few typos
>
> > Throughout, there are expressions of the form $\Omega=\prod_{i=1}^d x^{(k)_{k=1}^K}$ where $i$ is not used
>
> Thank you for pointing it out. Here, $i$ denotes an index of the data dimension (i.e., $i = 1,..., d$). As you mentioned, it is almost meaningless, so we change the notation to $\Omega = ( \\{  x^{(k)}   \\}_{k=1}^K )^d $ in the updated version for clarity.
>
>
> > 1. Comparability of the “likelihood” metric
>
> > Prior continuous-model papers typically report dequantized likelihoods (e.g., uniform  or learned dequantization), or use a categorical  at a positive . Here the bound integrates to  with a quantized fixed-point parameterization. It isn’t obvious this produces the same quantity other works report. It would help to clarify whether the contribution is primarily the bound, the trained model, or both.
>
> I believe your concern is whether the likelihood values from our QDPM are comparable to those from existing continuous models computed on dequantized data. The answer is yes, because all these values can be interpreted as lower bounds on the log-likelihood for the original quantized discrete data. For details, please refer to Section 4.1 of the updated manuscript. This is the same reason why the likelihood of autoregressive models that treat pixels as discrete distributions, such as PixelCNN, can be compared with existing continuous diffusion models.
>
>
> > (1a) Type of bound
>
> > Please state at a high level whether your BPD corresponds to:
> > (i) discrete likelihood (categorical over 256 bins per channel),
> > (ii) dequantized continuous likelihood, or
>
> Our BPD is discrete log-likelihood, and it is comparable to dequantized continuous log-likelihood, because the latter is also a lower bound of discrete log-likelihood as in Eq. (31) in the updated paper.
>
> > (1b) Logp → BPD conversion
>
> > Right now the text says you “report [the] upper bound calculated with Eq. (19)” but does not specify the unit conversion or definition used to go from log-likelihood to BPD. Assuming the logp is computed correctly, could you specify the exact computation you used to convert from your logp bound to your BPD bound?
>
> Thank you for your suggestion. Yes, our reported score is converted to the BPD scale by dividing the original Logp value with $d \log 2$. We clarify it in Section 5 in the updated version.
>
> > (1c) Evaluating other models with your bound, and evaluating your model with other bounds
>
> Our NLL bound in Eq. (24) is derived based on our specific parameterization in Eq. (17), so applying it to other models is not possible. On the other hand, Eq. (10) does not depend on the parameterization of the signal predictor $\hat{\mathbf{x}}_\theta$, so this can be computed for any diffusion-based models, and it corresponds to the likelihood values reported in the prior works (e.g., VDM). In addition, Eq. (19), the limit version of Eq. (10), does not depend on the parameterization either, so we can calculate it for any diffusion models. Interestingly, however, if we calculate Eq. (19) for non-quantization-aware models (e.g., VDM) , the NLL always diverges to infinity since the integrand diverges around $t = 0$.
> In the case of our QDPM, the signal prediction error shrinks to zero much faster than the denominator $t^3$, so the divergence can be avoided, as can be seen from Eq. (24).
>
>
> In fact, the divergence of Eq. (19) for non-quantization-aware models is very natural, since these models do not guarantee that $x_0$ converges to a quantized value, so the model probability for each quantized value is almost always zero, making the log-likelihood diverge. Therefore, to make the discrete likelihood finite for these models, setting $t_{min} > 0$ is essential.
>
>
> > 2. (Important!) Change-of-variables detail in Eq. (22)
>
> > Question 1 concerned defining the BPD (assuming logp computation is correct). This question is about the correctness of the logp computation itself. When moving from Eq. (20) to Eq. (22) with the substitution $u=e^{-1 / t^2}$, the weighting induced by $du/dt$ must be accounted for; otherwise the Monte-Carlo estimator could be biased and underestimate NLL. Could you please show the exact steps (including Jacobian factors) to justify the unweighted expectation over  in Eq. (22)?
>
> Yes, we explicitly take $du/dt$ into account for the derivation. To clarify it, we add more details about its derivation in Eqs. (22)-(24) in the updated version, so please refer to it.

---

> > ### Author Response · Authors · 2025-11-21
> > **Response to Reviewer Pzwu (Part 2)**
> >
> > > 3. “Theoretical lower bound” – Q1
> >
> > > The lower bound  appears to be the code-length of a uniform categorical over the 50k images divided by dimension. This is a property of the finite dataset, not the underlying data distribution. Please justify using this as a target for generative modeling and clarify whether your evaluation implicitly allows near-memorization (e.g., concentrating mass on quantized locations). Is it a theoretical lower bound on the 50k images?
> >
> > Thank you for your insightful comments. We mentioned the theoretical lower bound of the test NLL just for reference, and we were not to state that achieving that lower bound is the ideal goal of generative models; the ultimate goal is to model the underlying data distribution, not the empirical distribution of the test set. This lower bound is solely to represent the theoretical limit when evaluating on the test set.
> >
> > > 4. “Theoretical lower bound” – Q2
> >
> > > Why is the word “test” used here? For CIFAR there are 50k train images and 10k test images; for ImageNet, ~1.2M train and 50k test.
> > > Was there a mix-up? The other numbers reported in Table 1 usually refer to the CIFAR test set of 10,000 datapoints (even though FID is often reported on the training set). Which sets were your BPD numbers computed on? Why do we compare CIFAR test BPDs to a lower bound computed on 50k training points?
> >
> > Thank you for pointing it out. You are right. The size of the CIFAR-10 test set is 10k (not 50k), so we modified the value in the updated version. The theoretical lower bound for the CIFAR-10 test set is $\log_2 10000 / (3 \cdot 32^2) \approx 0.0043$.
> >
> > > 5. Definition of $s_t$
> >
> > Thank you for your suggestion. We have added the definition of $s_t$ in the updated version.
> >
> > > 6. Code availability
> >
> > > The anonymous code link in Appx. B.1 is currently empty. It would be helpful to share working code so others can confirm the BPD computation protocol.
> >
> > Sorry for the inconvenience. We have uploaded the code to the provided link, so please refer to it.
> >
> > > 7. Sensitivity of reported BPD
> >
> > > How sensitive are the reported BPD values to the time-discretization schedule? In my experience, sampling 1,000 vs. 10,000 uniform points in (0,1) can drastically affect dequantized-diffusion BPDs for standard noise schedules and common checkpoints (e.g., those from Song, Durkan & Song, Kingma, etc.).
> >
> > The BPD for our QDPM is calculated using Eq. (24) with 30k uniform samples of $u$, and we confirm that the standard deviation for five random seeds is sufficiently small ($\ll 0.01$).
> >
> > Feel free to ask further questions if you may have.
> >
> > Thanks.

---

> ### Author Response · Authors · 2025-11-28
> **Gentle Reminder to Reviewer Pzwu**
>
> Thank you again to your efforts in reviewing our paper and your constructive comments. This is a gentle reminder that we have updated the paper to address your concerns. The discussion period will end in five days, so please let us know if you have further comments about the update.
>
> Thanks!

---

### Official Review · Reviewer_poSH · 2025-11-11

**Soundness:** 3
**Presentation:** 3
**Contribution:** 2
**Rating:** 4
**Confidence:** 4

**Summary:**

- This paper introduces Quantizing Diffusion Probabilistic Models (QDPM), a diffusion model that explicitly integrates quantization into the  reverse diffusion process.
 - Unlike traditional models that treat data as continuous or rely on dequantization noise, QDPM enforces convergence to quantized points through a carefully designed signal predictor.
 - The authors prove theoretical guarantees, derive a quantization-aware maximum likelihood objective, and develop efficient SDE solvers.
 - Experiments on CIFAR-10 and ImageNet-32 show remarkable improvements in negative log-likelihood (NLL), from prior SoTA of 2.42 bpd to 0.27 bpd, nearing the theoretical lower bound.

**Strengths:**

1. The paper provides a clear theoretical foundation for incorporating quantization into the reverse SDE. The use of softround + decaying correction (Eq. 17) is interpretable.
2. The analysis of convergence (Propositions 1 and 2) and the closed-form derivation of the SDE solution demonstrate theoretical grounding.
3. Achieving a 0.27 bpd test NLL on CIFAR-10 is a good improvement and suggests genuine progress in likelihood-based training of diffusion models.
4. The proposed QDPM framework could influence how quantization is treated in generative modeling, especially for discrete data domains (e.g., images, audio).

**Weaknesses:**

1. The paper emphasizes log-likelihood but gives limited discussion on sample quality (FID ≈ 5.6 / 8.9). While the authors claim NLL–FID trade-off is expected, a deeper analysis (e.g., perceptual vs likelihood trade-off curve) would strengthen claims.
2. Although the authors cite dequantization-based diffusion models, there are no direct experimental comparisons using the same architecture but with traditional dequantization. This would isolate the quantization-aware effect.
3. The paper reports approaching the theoretical lower bound (~0.0051 bpd), but this bound assumes uniform discrete entropy over the test set; a short derivation or clarification would help readers assess the significance.
4. There are no ablations on components such as the softround function, or solver order. These would be important to understand where the gain originates.

**Questions:**

Please check the weaknesses and answer the questions.

---

> ### Author Response · Authors · 2025-11-21
> **Response to Reviewer poSH**
>
> Thank you for your thoughtful review. We listed our responses to address your concerns below.
>
> > The paper emphasizes log-likelihood but gives limited discussion on sample quality (FID ≈ 5.6 / 8.9). While the authors claim NLL–FID trade-off is expected, a deeper analysis (e.g., perceptual vs likelihood trade-off curve) would strengthen claims.
>
> Thank you for your insightful suggestion. To make the trade-off between likelihood and perceptual quality, we have added Figure 6, plotting the negative log-likelihood (NLL) and FID of our QDPM and other existing diffusion models, including maximum-likelihood models (VDM and ScoreFlow) and non-likelihood-based models (Improved DDPM, DDPM++ and NCSN++). As can be seen from it, models trained with likelihood loss tend to achieve better NLL, while their FID tends to be worse than non-likelihood models, showing the trade-off between them.
>
> > Although the authors cite dequantization-based diffusion models, there are no direct experimental comparisons using the same architecture but with traditional dequantization. This would isolate the quantization-aware effect.
>
> We use the same architecture with VDM as described in the first paragraph of Section 5, so the direct comparison of the quantization-aware effect can be obtained by comparing VDM and our QDPM. In VDM, quantization is taken into account by defining $p ( x_0 \mid x_{t_{min}} )$ as a discrete categorical distribution. To compare with an explicit dequantization-based method, we have also added a result of the case, where VDM is trained with uniform dequantization, in Table 1 in the updated version. As can be seen from these comparisons, the drastic likelihood improvement directly comes from our proposed quantization-aware parameterization.
>
> > The paper reports approaching the theoretical lower bound (~0.0051 bpd), but this bound assumes uniform discrete entropy over the test set; a short derivation or clarification would help readers assess the significance.
>
> Thank you for your insightful suggestion. We have added an explanation on the derivation of the theoretical lower bound in Appendix C. In a nutshell, the test likelihood is maximized when the model distribution corresponds to the empirical distribution of the test set, so assuming that there is no overlap between data samples inside the test set, the theoretical lower bound of the NLL corresponds to uniform discrete entropy over the test set.
>
> > There are no ablations on components such as the softround function, or solver order. These would be important to understand where the gain originates.
>
> For our QDPM, we can evaluate the upper bound of NLL using Eq. (24) of the updated paper without simulating the reverse SDE, so the quantitative result in Table 1 does not depend on the choice of the SDE solver.
> Therefore, we can investigate the source of performance gain by purely comparing QDPM with VDM, which shares the experimental design except for the formulation of the signal predictor in Eq. (14).
>
> Feel free to ask further questions if you may have.
>
> Thanks.

---

> > ### Author Response · Authors · 2025-11-28
> > **Gentle Reminder to Reviewer poSH**
> >
> > Thank you again to your efforts in reviewing our paper and your constructive comments. This is a gentle reminder that we have updated the paper to address your concerns. The discussion period will end in five days, so please let us know if you have further comments about the update.
> >
> > Thanks!

---

### Author Response · Authors · 2025-12-03
**Final Remarks by Authors**

Dear Area Chairs and Reviewers,

Thank you for your effort during the review and rebuttal process, especially given the unusual situation. We truly appreciate your work in keeping the review process constructive.

We conclude our rebuttal with a summary of the major updates made in response to the reviewers’ comments:

**Derivations of theoretical NLL lower bound**

To answer the questions from Reviewer poSH and Reviewer Pzwu, We have added an explanation on the derivation of the theoretical lower bound in Appendix C.

**Justification of comparison between discrete NLL and dequantized NLL**

Reviewer Pzwu had a concern about whether the comparison between NLL for discrete data and dequantized data. To make it clear, we have added an explanation why the comparison between them is valid in Section 4.1. In a nutshell, the continuous log-likelihood value for dequantized data can be regarded as lower bounds on the log-likelihood for the original quantized discrete data. Such comparison is commonly performed when we compare the likelihood of discrete autoregressive models (e.g., PixelCNN) and continuous models (e.g., diffusion).

**Change-of-variables detail in Eq. (22)**

To address the concern by Reviewer Pzwu, we have added a derivation of our log-likelihood loss in Eq. (22)-(24), clarifying the change-of-variable procedure.

**Empirical validation of Proposition 1**

We have added an experiment to verify that Proposition 1 holds in practical situations, in which we measure the fixed-point error at each time step when sampling from the pretrained EDM on CIFAR-10 and AFHQ and plot it in Figure 3. As can be seen from it, the fixed-point error converges to zero as $t \to 0$. We believe that this result can address the concern by Reviewer Zbvq.

**Relation to EqM**

To answer the question from Reviewer 12ST, we have added a discussion on the relation to the equilibrium matching model (EqM) in Appendix B.

**Code availability**

We have updated the anonymized public codebase in Appendix D.1 for the reproductivity, which was empty at the time of submission.

**Fixing typos**

We have fixed all the typos pointed out by the reviewers in the updated version. We have also added the definition of $s_t$ based on the suggestion by Reviewer Pzwu.

We believe we have addressed all of the reviewers’ concerns and sincerely appreciate your consideration.

Best regards, Authors of Submission 7574

---

### Meta-Review · Area_Chair_nare · 2025-12-21

**Summary:**

The paper proposes a quantization-aware diffusion parameterization that provably converges to discrete pixel values, enabling valid discrete likelihood evaluation and achieving a striking NLL improvement (0.27 bpd on CIFAR-10), near the dataset’s lower bound.

Pros
* Clear, principled treatment of quantization in diffusion models
* Strong theoretical guarantees with careful likelihood derivations
* Large, convincing NLL gains over prior work
* Rebuttal thoroughly addresses reviewer concerns (derivations, bounds, comparisons, code)

Cons
* Expected likelihood–FID trade-off; sample quality lags non-likelihood models
* “Theoretical lower bound” is dataset-dependent and requires careful interpretation

Overall, a solid, well-justified contribution that meaningfully advances likelihood-based diffusion modeling for discrete data.

**Reviewer Scores:**

n/a

---

### Decision · Program_Chairs · 2026-01-26

Accept (Poster)